# Evaluation and Optimization Model of Rural Settlement Habitability in the Upper Reaches of the Minjiang River, China

**DOI:** 10.3390/ijerph192214712

**Published:** 2022-11-09

**Authors:** Hao Mei, Jin Yang, Mingshun Xiang, Xiaofeng Yang, Chunjian Wang, Wenheng Li, Suhua Yang

**Affiliations:** 1College of Tourism and Urban-Rural Planning, Chengdu University of Technology, Chengdu 610059, China; 2Research Center for Human Geography of Tibetan Plateau and Its Eastern Slope, Chengdu University of Technology, Chengdu 610059, China

**Keywords:** rural settlements, livability evaluation, optimization mode, rural revitalization, the upper reaches of the Minjiang River

## Abstract

Rural settlements are the spatial carriers of rural multifunctionality, and various issues related to livability are the main manifestations and causes of unbalanced and insufficient rural development. In the new era, it is imperative to promote the livability of rural settlements with the implementation of rural revitalization. However, compared with urban settlements, there are still fewer studies on the livability of rural settlements, especially those in disaster-prone areas; thus, this paper takes the upper reaches of the Minjiang River as the study area. It adopts GIS spatial analysis and the model of minimum cumulative resistance, etc., to conduct a livability evaluation and construct an optimization model by innovatively taking five aspects into account including site security and resource endowment. The results show that: (1) The overall livability of the region is relatively good, and the main factors affecting the livability are site security and economic affluence; (2) The location of rural settlements was highly livability-oriented, and the area of rural settlements in the moderate- and high-livability zones accounted for more than 90%; and (3) The key to improving the livability of rural settlements lies in the construction of development synergy, disaster management, cultural preservation and industrial upgrading, and thus, four types of settlement livability enhancement are proposed. The research results provide theoretical support for the construction of livable villages in the upper reaches of the Minjiang River and similar mountainous areas.

## 1. Introduction

With the acceleration of urbanization in the world, rural decline is a global issue [1]. Since the implementation of the reform and opening-up policies in 1978, China has witnessed rapid urban development, and “rural issues” such as inadequate infrastructure, inaccessible public services, unsatisfactory living conditions, abandoned farmland, population loss and ageing have become increasingly prominent [2,3]. In retrospect, the implementation of policies to solve issues relating to agriculture, rural areas, and rural people, including new rural construction, targeted poverty alleviation, rural revitalization and territorial planning were aimed at addressing major issues such as unbalanced and inadequate development [4]. At the same time, with the in-depth study of human geography in recent years, international research on rural settlement has gradually turned to rural reconstruction, rural social problems and other aspects [5]. In the new era, the livability of rural settlements has a profound impact on the implementation of rural revitalization. Therefore, constructing a scientific index system for evaluating the livability of rural settlements and proposing reasonable livability enhancement strategies are important entry points for achieving coordinated development, and have important theoretical and practical significance for promoting the implementation of rural revitalization.

In recent years, scholars’ evaluation of livability can be divided into two categories: one is the comprehensive evaluation of livability based on multiple factors and dimensions [6] and the other is research on certain indicators of livability, including safety livability, ecological livability and building livability [7,8]. The habitability evaluation methods commonly used in domestic and international research include the entropy weight method, the analytic hierarchy process, the fuzzy synthesis estimate method, the neural network method, the matter element extension method, the principal component analysis method, the network hierarchy method, and the exploratory structural equation model (ESEM) [9,10,11,12]. In terms of livability in urban areas, the World Health Organization (WHO) proposed the basic living standard in 1961, and indicated that basic human life should be safe, healthy, convenient, and comfortable [13]. This metric was widely used in the evaluation of urban livability. Zhang [14] believes a livable environment contains both an attractive, clean and harmonious natural ecological environment and a safe, convenient and comfortable social and humanistic environment. Wang, et al., [15] introduced urban livability as a “factor” into the research framework of the production function from a macro point of view, constructed a theoretical model of the impact of economic development on urban livability, and explored the impact of urban economic development on its livability. Conversely, Huang, et al. [16], from a micro point of view, assessed community livability in major cities in China from the perspective of remote sensing and volunteered geographic information. In the study of the livability of rural settlements, the precondition of carrying out the optimization of habitable rural settlements is conducive to knowing the habitability of rural settlements at the present stage and interpreting the spatial differentiation characteristics of rural settlement habitability. This allows us to implement the differential optimization regulation strategy of rural settlement habitability [17].

In order to address the problems of construction land expansion encroaching on surrounding villages, the scattered layout of rural settlements, and low efficiency of rural land use, relevant scholars often classify rural settlements into different upgrading types and propose corresponding optimization and upgrading strategies. The optimization of rural settlements can take two directions: one is to optimize the spatial hierarchical relationship of settlements and their governance system to form a clear hierarchical functional positioning [18,19,20]; the other is to explore the reasonable regulation and coordination of regional tertiary and biological space to identify various patterns of settlement development [21,22,23]. Long and Liu [24] divided the types of rural settlement reconstruction into three aspects, namely, spatial structure, economic structure and social structure reconstruction, which emphasized the key points of rural structural adjustment in China. Bi, et al., [25] proposed the reconstruction framework of rural settlements with an eye to regional ecology, production, and livelihood. However, Zhong, et al., reveled that geological hazards were the main driving force of mountainous settlement reconstruction by combining PRA with the structural equation model [26]. 

Currently, although there are abundant research results on the evaluation and optimization of settlement livability, there are few research results on rural settlements, especially those in areas prone to geological disasters. The upper reaches of the Minjiang River is ecologically fragile and is the site of frequently occurring geologic calamities. It is typical of a socio-economically underdeveloped area of China, and implementing rural revitalization there is difficult. As a result, the composite factors lead to a more complex evolution of rural settlements and more prominent livability problems in this area. Therefore, this study takes the upper reaches of the Minjiang river as the study area by considering various factors like the natural resources, economic development, and the condition of sociability. In addition, the livability evaluation index system is innovatively constructed from the aspects of security, resources and convenience by spatializing the per capita resource quantity factor. This study also takes the current settlement map as the statistical unit and uses GIS spatial analysis technology to achieve the following objectives: (1) To build the evaluation index system of rural settlement livability; (2) To study the habitability of rural settlements in the study area; and (3) To propose a suitable livable settlement optimization model, and to zone the optimization model by village. This study contributes to optimizing the theoretical framework of the study on the evaluation and improvement strategies of habitability in disaster-prone areas and can provide a clear idea and scientific basis for the improvement of rural settlement livability in the upper reaches of the Minjiang River and similar alpine valleys and disaster-prone areas in the world.

## 2. Materials and Methods

### 2.1. Study Area

The upper reaches of the Minjiang River are in the intersection of the Qinghai-Tibet Plateau and the mountainous area of the western Sichuan Basin, with a geographical position between 30°45′ N~33°10′ N and 102°59′ E~104°14′ E (Figure 1). It borders the Sichuan Basin in the east, the eastern margin of the Qinghai-Tibet Plateau in the west, and the Zoige Plateau in the northwest. The mean altitude in the region is over 3300 m, and it has a surface area of 2.48 × 10^4^ km^2^. There are 17,492 rural settlement patches in the region, with a total area of 58.2298 km^2^. This region belongs to the plateau alpine monsoon climate, which, like the monsoon climate, is divided into dry and wet seasons. The rainy season lasts from May to September, which is a period of high humidity that accounts for over 60% of the rainfall for the whole year [27]. The other seasons have little rain, with an annual rainfall of 500–850 mm [28]. The soil types mainly include subalpine meadow soil, cold desert, stony soil, and brown soil [29]. The vegetation types include green deciduous broad-leaved mixed forest, coniferous broad-leaved mixed forest, coniferous forest, shrub and grassland. Thus, this region is biodiversity-rich [30,31]. As a result of the special geographical environment and customs, there are a large number of unique Tibetan and Qiang settlements in this area, which is an important reserve of Tibetan and Qiang culture in China. This area is an important ecologically sensitive area and ecological environment protection zone in China because of frequent natural disasters, which pose threats to the safety of rural settlements and the productivity and lives of residents.

### 2.2. Research Data

The geological fault lines are from the China Earthquake Fault Information System, Institute of Geology, China Earthquake Administration. The soil organic carbon (SOC) data are obtained from the National Data Center for Glaciology, Permafrost and Desert Science. In addition, the Van Bemmelen factor was used to convert the SOC data into the soil organic matter data [32]. The National Earth System Science Data Center provides the near-surface mean air temperature, relative humidity, mean wind speed and PM2.5 concentration. The strength of this dataset lies in that it can reflect the small differences of meteorological conditions within a region caused by special regional topography and hydrological conditions, and minimizes errors caused by spatial interpolation. See Table 1 for specific data sources.

### 2.3. Research Framework and Index System

#### 2.3.1. Research Framework

This study was carried out by following the procedures of data preparation and preprocessing, livability evaluation, and livability improvement of rural settlements (Figure 2). The data preparation stage mainly includes the acquisition and processing of multi-source data. In the comprehensive evaluation stage, the livability index is spatialized and standardized, and the comprehensive index of rural settlement livability in the upper reaches of the Minjiang River is obtained by GIS operation. Lastly, based on the results and field research, the paper puts forward the model and strategy of improving the livability of rural settlements in the upper reaches of the Minjiang River.

#### 2.3.2. Evaluation Index System of Rural Settlement Livability

In the existing research results, the selection of livability evaluation indexes mainly considers the regional natural environment and economic access in terms of geology, topography, transportation, natural resources, social economy, pollution and other aspects [33,34,35]. Based on previous studies, this paper has different priorities in the selection of indicators. Because of the abrupt and intermittent indicators that may affect the overall rationality of the evaluation results of livability, 25 secondary indexes were finally selected by considering five aspects including site security, resource endowment, environment suitability, accessibility of life, and affluence of economy to build the evaluation index system of livability. As the problem of safe drinking water in the study area has been addressed, the water resources issue has been classified into an endowment. In addition, efforts were made to build the evaluation index system of rural settlements habitability, which is shown in Table 2.

### 2.4. Methods

By referring to the relevant research results [37,47,48], the accuracy of data resampling is ensured, and the spatial differences of evaluation factors in the region are reflected through practices to improve the overall accuracy of the study. This paper uniformly resamples the acquired data to the grid unit of 500 m × 500 m in ArcGIS 10.2 platform, and the projection coordinate system is unified as “WGS_1984_UTM_Zone_48N”. Among those, Distance from Fault, Distance from Factories, and Distance from Counties are obtained based on Euclidean distance analysis in ArcGIS platform. All secondary indexes were preprocessed and normalized. Finally, the spatial superposition method of grid data is used to convert each evaluation index into the same spatial base, and the livability index corresponding to each grid is obtained with a map algebra operation. The main indicators and livability index are calculated as follows.

#### 2.4.1. Kernel Densitometry Analysis

Rosenblatt proposed the kernel density analysis method, which is widely used to analyze the spatial agglomeration degree [38]. The density of geological disaster points in the index system is calculated by this method, and the formula is as follows:(1)$$f_{h}\left(x\right)=\frac{1}{nh}\overset{n}{\underset{i=1}{\sum }}K\left(\frac{x-x_{i}}{h}\right)$$
in Formula (1), $f_{h}\left(x\right)$ is the kernel density; $x_{i}$ represents each disaster point; x − $x_{i}$ indicates the distance between the estimated points $x_{i}$ and the known points, while h is bandwidth. K is the Gaussian kernel function.

#### 2.4.2. Degree of Topographic Relief

Topographic relief is a key factor affecting regional rural settlement distribution, population size and economic activities, as well as an essential element influencing soil erosion and soil fertility loss. The formula is as follows [49]:(2)$$R_{\mathrm{DLS}}=\frac{\left[\mathrm{Max}\left(H\right)-\mathrm{Min}\left(H\right)\right]\times \left[1-P\left(A\right)/A\right]}{500}$$

In Formula (2), $R_{\mathrm{DLS}}$ presents the degree of topographic relief; Max(H) − Min(H) is the relative elevation difference; P(A) and A are the area and total area of the flat area, respectively. The area with slope less than or equal to 5° is considered as the flat area.

#### 2.4.3. Per Capita Land Area and Density of Water and Road Network

The data of woodland, grassland, cultivated land, road and water systems are extracted, respectively. Additionally, a 500 × 500 m grid is created. The number of people, the area of various land types (km^2^) and the length of road networks and water networks (km) are counted in the grid, and the per capita area of each type of land and the density of road networks and water networks are calculated. In light of the fact that the population in the grid is less than one person in the statistical results, the grid with less than 1 person is defined as 1 person to ensure the reliability of the calculation results, and the formula is as follows:(3)$$P_{\mathrm{ij}}=\frac{S_{\mathrm{ij}}}{{\mathrm{pop}}_{\mathrm{ij}}}$$
(4)$$D_{j}=\frac{L_{j}}{S_{j}}$$
in Formula (3), P_ij_ is the per capita occupancy of type I land in grid J (km^2^/person); S_ij_ presents the area occupied by type I land in grid J and pop_ij_ represents the population number in grid J. In Formula (4), D_j_ means the density of road network/water network in grid J (km/km^2^); L_j_ is the length of road network/water network in grid J and S_j_ presnets the area of the J grid.

#### 2.4.4. Meteorological Indices

The comfort level [41,42,50] is defined by the temperature and humidity index and wind efficiency index in the comfort evaluation of human settlements, and the formula is as follows:(5)$$T_{hi}=\left(1.8t+32\right)-0.55\left(1-f\right)\left(1.8t-26\right)$$
(6)$$W_{\mathrm{ei}}=8.55S-\left(10\sqrt{V}+10.45-V\right)\left(33-t\right)$$
in Formulas (5) and (6), $T_{hi}$ means temperature and humidity index; $W_{\mathrm{ei}}$ presents wind effect index, and t, f, S and V represent the average temperature (°C), relative humidity (%), sunshine duration (h) and wind speed (m/s), respectively.

#### 2.4.5. Resistance Value of Accessibility

By referring to the method in calculating the resistance value of road accessibility, and combining it with the minimum cumulative resistance model used by Knaapen [51], efforts are made to construct the resistance value index of the roads [43], schools [44] and general hospitals [45] to reflect the convenience of life in the upper reaches of the Minjiang River. Then, the minimum cumulative resistance value of each grid to the destination is calculated as follows:(7)$$M_{\mathrm{CR}}=f_{\min}\sum \left(D_{\mathrm{ij}}\times R_{j}\right)$$

In Formula (7), M_CR_ means the minimum cumulative resistance value; D_ij_ represents the spatial distance of the object from source j to landscape unit I and R_j_ presents the resistance value of landscape element j.

#### 2.4.6. Index Weight

The AHP-Entropy method is used to determine the index weight [52,53]. The analytic hierarchy process (AHP) is a systematic analysis method combining qualitative and quantitative analysis [54], which can be used to determine the subjective weight ${W^{\prime }}_{j}$ of each index. The entropy method determines the index weight according to the original information of the index item, which not only reflects the effect value of the index information, but also avoids the information overlap among the indicators. It is a relatively objective multi-index evaluation method [39], and is used to determine the objective weight of each indicator ${W^{\prime \prime }}_{j}$. On this basis, the combined weight of the evaluation index of rural settlement livability is obtained: $W_{j}$. The distance function [55] is introduced to eliminate the interference of large fluctuations of data, so that the difference degree between subjective and objective weights is consistent with the difference degree of combination coefficients α and β. The formula is as follows:(8)$$W_{j}=\alpha {W^{\prime }}_{j}+\beta {W^{\prime \prime }}_{j}$$
(9)$$D({W^{\prime }}_{j},{W^{\prime \prime }}_{j})=\sqrt{\frac{1}{2}\sum_{i=1}^{n}{\left({W^{\prime }}_{j}-{W^{\prime \prime }}_{j}\right)}^{2}}$$

In the formula, α and β are mainly the combination coefficients of objective weights, and the sum of them is 1, while D(${W^{\prime }}_{j}$, ${W^{\prime \prime }}_{j}$) is the distance function between the main objective weights. Additionally, α and β are 0.6074 and 0.3926, respectively, and the weight calculation results are shown in Table 2.

Then, the weighted sum calculation is carried out to obtain the function evaluation index of each system, $F_{j}$, and the formula is:(10)$$F_{j}=\overset{m}{\underset{i=1}{\sum }}W_{j}Y_{\mathrm{ij}}$$

## 3. Results

### 3.1. Livability Zoning

In line with the single factors and weights in Table 2 and the multi-factor weighted evaluation model, the map algebra method is used to calculate the single livability index of each grid cell. The natural break point method is used to classify the evaluation results into five grades: unsuitable, relatively unsuitable, low suitable, moderately suitable and highly suitable according to the livability index from low to high (Figure 3).

From the spatial pattern of each livability grade, the upper reaches of the Minjiang River are generally poor in terms of the site security, resource endowment and economic affluence, while the environmental suitability and living convenience are relatively better. Site security is affected by regional geology and topography. The area where site safety is moderate and high is mainly distributed in Heishui County, the eastern river valley in Maoxian County and the western regions in Wenchuan County, while the site insecurity zone is mainly distributed in the eastern and southwestern regions of the study area, especially in the valley area. In terms of resource endowment, there is a distinct north and south divide of the study area. Additionally, the north is significantly better off than the south, which is due to the high per capita availability of woodland, arable land and grassland resources in the north. Concurrently, the soil fertility in the north is higher and solar energy resources are more abundant. Environmental suitability and convenience of living have similar characteristics, and the endowment degree decreases from the valley region to the periphery, which is related to the more-livable climate environment in the valley region. The convenience of transportation, medical care, education, and employment in the river valley is also better than in other areas. In terms of economic affluence, the moderately and highly livable areas are mainly distributed around areas that are heavily influenced by human activities, with dense economic activities and population. These areas have more developed industry and commerce, more employment opportunities and a higher degree of convenience of life.

The comprehensive livability is divided into five levels based on the natural breakpoint method, and the area of each level is counted, as shown in Figure 4.

Among the study area, the area of low livable area is the largest, accounting for 34.26%, while the area of unlivable area is the smallest, accounting for 8.03%. Specifically, in order of descending amounts of geographical area, the area that is low livable area > moderate livable area > relatively unlivable area > highly livable area > unlivable area, indicating that the comprehensive livability of the study area is generally high, which is the main reason why there are still a large number of rural settlements distributed in the area despite the frequent occurrence of natural disasters. From the perspective of spatial distribution, highly livable areas are mainly distributed in and around the county with frequent economic activities and well-developed public service facilities, with obvious distribution characteristics along transportation lines and water systems. The moderately livable areas are located around the periphery of the highly livable areas, mainly in the mid-hill and high- mid-hill areas. The low livable areas are mainly located at the edge of the moderately livable areas, which is a cross-transition region of habitability status and a potential zone of habitability improvement. The relatively unlivable and unlivable areas are mostly located in areas with high altitude, low living convenience and poor climatic conditions, where human construction activities should be avoided to prevent the deterioration of the ecological environment.

### 3.2. Livability Analysis of Rural Settlements

There are 17,492 rural settlements in the upper reaches of the Minjiang River, with a total area of 58.2298 km^2^. The rural settlement pattern was used as the statistical unit to analyze the livability of each rural settlement (Table 3).

In short, the area of rural settlements in the upper reaches of the Minjiang River decreased with the reduction of habitability. In addition, the areas with moderate and high habitability are the core areas for residents to live in this region. Among them, 53.844% of the rural settlements are highly livable; 36.829% are moderately livable; 9.309% are lowly livable; and 0.018% are relatively unlivable. There are no unlivable rural settlements. In terms of the spatial distribution of rural settlements with different livability, the counties, key towns, and their surrounding areas in the river valley area have higher livability, mainly because these areas have frequent economic activities, are close to all amenities, and are safe and environmentally friendly, such as Pingtou Village, Dongyu Village and Shuanghe Village. There are also some rural settlements with good living conditions in the mid-mountain and high-mid-mountain areas, mainly due to the high degree of resource endowment, environmental suitability, and life convenience, such as Bingli Village and Ximending Village. However, the rural settlements with lower livability are mainly those with low economic affluence, resource endowment and site security, such as Aer Village, Longtan Village and Sancha Village.

As displayed in Figure 5, the average livability index within the buffer zone of 100~2000 m decreases from 0.0784 to 0.0705, generally showing a trend of a lower livability index as the distance from the rural settlement increases. Considering the spatial relationship between rural settlements and unsuitable areas, there are a total of 84 settlements within 500 m of unsuitable areas, 199 settlements within 1000 m and 387 settlements within 1500 m. There are no settlement points within 500 m of the unsuitable area, 3 settlement points within 1000 m and 13 settlement points within 1500 m. All of the above indicated that most of the current rural settlements in the upper reaches of the Minjiang River have been in the relatively best position in the region through implementing the policies of poverty alleviation, relocating from uninhabitable areas, project relocation in recent years, and the continuous improvement of public infrastructure. From now on, the improvement of the livability of rural settlements in the region should focus on individual livability indicators, such as the improvement of the security of settlement sites, transportation service facilities, and the multi-functional utilization of resources, etc., without the need to carry out large-scale relocation projects.

### 3.3. Optimization Model of Rural Settlements Livability

Five major issues have surfaced as a result of our study, namely, site security, resource endowment, suitable environment, life accessibility and economic prosperity. These areas continue to impede the improvement of the livability of rural settlements in the upper reaches of the Minjiang River. Firstly, rural settlements near counties and towns are often inhabitable, but the internal force of development is insufficient. Secondly, some scattered rural settlements are too far away from town centers, and most of them are inhabited by the aged or have been directly abandoned, such as Sandagu Village in Heishui County. Thirdly, the security of some rural settlements is not guaranteed. That is to say, there are potential security risks, such as are seen in Baibu Village in Maoxian County and Zhangpai Village in Wenchuan County. Fourthly, the overall situation of resources is poor and underutilized, and the industrial support for development is insufficient. Fifthly, the lack of preservation and excavation of cultural heritage in some rural settlements leads to the loss of cultural characteristics. Aiming at settling the situation, this study proposes the optimization model (Figure 6) of rural settlements in the upper reaches of the Minjiang River based on four optimization objectives including development synergy, disaster management, industrial upgrading and cultural preservation. 

Type 1. Clustering Improvement. This upgrading type mainly involves rural settlements around the counties and core towns, which are highly suitable areas for livability, such as Zongqu Village, Nanzhuang Village, Jingzhou Village and Tai’an Village in Maoxian County. Because they are close to the town center, they have a high degree of convenience and economic affluence, good traffic conditions, high development potential, and can be driven to improvement by the development of the regional economic center. In line with the principle of “building central villages and towns and making intensive and efficient use of land resources”, these settlements will further promote new urbanization, guide the population to gather in towns and central villages, and build a new urban–rural development pattern. Through policy guidance, improving education, medical facilities and other basic services in the central village, exploring suitable industrial development directions to create employment opportunities, attracting the surrounding residents to gather in the central village and town and gradually building a large-scale central village and town, the problem of scattered land use and the low level of intensification in the district can effectively be solved. Meanwhile, the driving role of the central village and town in the development of surrounding areas should be clarified, forming a linkage development with the town center, and driving the overall progress of the region.

Type 2. Restructuring and Disaster Prevention. Following the optimization idea of “policy support, disaster prevention and avoidance, and ecological relocation”, this type of upgrading mainly focuses on the settlements with low site security and poor livability. On the one hand, for the settlements with high security risks such as Baibu Village, Sujiaping Village, Bazhu Village in Maoxian County and Zhangpai Village in Wenchuan County, etc., reasonable engineering measures should be considered first, and at the same time, the site security of village settlements should be improved through preventive and early warning measures and protective treatment. If the site is unsafe, historical disasters are frequent and uncontrollable, resources are insufficient and the convenience of living is poor, we should organically combine disaster prevention and avoidance with ecological environment restoration, formulate incentives for population migration, and carry out the appropriate relocation. On the other hand, for areas that are too far away from the central village, too scattered in distribution and with low production convenience, such as the western part of Sandagu Village in Heishui County, the southwestern part of Wolongguan Village in Wenchuan County and the northwestern part of Lianghekou Village in Songpan County, which are few and scattered and are mostly inhabited by the elderly population, or have been directly abandoned, residents in these area should be guided to relocate to the advantageous areas in a reasonable, moderate, gradual and orderly way under the dual principles of following the rules of village development and respecting the will of the residents.

Type 3. Agriculture and Animal Husbandry Development. Adhering to the development orientation of “greening and branding”, we actively explore the development of regional specialism in agriculture and animal husbandry, mainly involving rural settlements with high environmental suitability, resource endowment and convenience of life. On the one hand, in the relatively flat, concentrated and easier-to-cultivate areas of arable land, such as the village of Renentang in Heshui County, the multi-functional utilization performance of land can be enhanced by creating special ecological agriculture and animal husbandry industries in patches or bands according to local conditions, promoting the integrated development of agriculture, animal husbandry and secondary and tertiary industries and strengthening the brand and quality of construction of agricultural products. Building plateau agriculture and animal husbandry bases with good economic benefits will have strong income-generating effects. On the other hand, in areas with relatively poor farming conditions, farmers can be guided to make reasonable use of the front and back yards of their houses to explore development models such as “micro pastoral” and “micro pastoral + tourism”, forming the business model of “company/cooperative + base + farmers”, promoting the organic combination of modern civilization and rural features, optimizing the use of land and enhancing the interest of farmers’ lives.

Type 4. Integrated Development of Culture and Tourism. Intensive efforts should be made to adhere to and promote the “Tibetan and Qiang cultural deposits” and blaze a trail of an “ecological cultural tourism brand”. This enhancement type mainly involves villages inhabited by Tibetan and Qiang people that have a suitable environment and convenient transportation. These villages should be built on the basis of preserving and excavating regional cultural resources, creating ecological and cultural tourism brands, and constructing a regional characteristic tourism development pattern. On the one hand, in the construction of village scenic spots and the integrated development of culture and tourism in villages with Tibetan and Qiang characteristics (for example, Ganbao Village, Luobozhai Village, etc.), development should focus on the protection and inheritance of Tibetan and Qiang culture, fully explore and retain the traditional ethnic customs and cultural characteristics, and promote the declaration of cultural heritage of ancient architecture, culture, crafts, etc. In addition, while retaining the original architectural style in the village renovation, special cultural experience projects should be added to drive tourists to understand and perceive the ethnic culture, forming a unique attraction of Tibetan and Qiang culture. On the other hand, tourism resources such as Tibetan and Qiang ethnic cultures, the spirit of the Long March, and natural scenery should be integrated, and tourism infrastructure should be upgraded to promote the integrated development of “transportation + tourism”. In this context, major progress will be made in building red and natural scenery tourism lines. By relying on the existing scenic spots, efforts should be made to create a new tourism marketing mode, increase the key scenic spots, enhance the market influence of cultural and tourism brands, establish tourism promotion alliances, and build regional representative cultural and tourism brands.

Based on the above analysis and livability optimization model, the village level administrative unit is taken as the research scale to divide the livability optimization model of rural settlements in the upper reaches of the Minjiang River. The results are shown in Figure 7.

## 4. Discussion

This paper adopts the grid data of space superposition method of each evaluation index into the same space base though map algebra operations to obtain the habitability of the corresponding index of each grid. This evaluation has identified the main issues of improving the livability of rural settlements in the upper reaches of the Minjiang River. Four specific optimization models to upgrade the livability of rural settlements are proposed considering synergistic development, disaster management, industrial upgrading and preserving and promoting cultural heritage as the main optimization objectives. The results can provide a clear scientific basis for the improvement of rural settlement livability in the upper reaches of the Minjiang River and similar alpine valleys and disaster-prone areas in the world and optimize the theoretical framework for the evaluation of the livability of rural settlements in disaster-prone areas and the study of the improvement strategy.

The research results indicate that, from the livability classification results of the upper reaches of the Minjiang River, the environmental suitability and living convenience are relatively good, but, at the same time, this region is poor regarding site security, resource endowment and economic development, which is consistent with the fact that the region is a geologically disaster-prone and underdeveloped area [56,57]. Simultaneously, the relevant studies show that the overall risk of mountain disasters is higher in the upper reaches of the Minjiang River, and the risk of mountain disasters is higher in the eastern and southern regions of Maoxian, Lixian and Wenchuan Counties [58]. With the acceleration of urbanization in recent years, the region has been optimized in terms of infrastructure construction, population increase, and the accessibility of hospitals and education; the convenience of life has been improved. The areas with better convenience of life are mainly located on both sides of the road, which is consistent with the research conclusion of Wang, et al. [59].

Judging from the habitable status of rural settlements in the upper reaches of the Minjiang River, the area of plaque in rural areas decreases as the habitability reduces. The moderately and highly habitable areas are the core areas of human settlement in this area. The results show that through the policy of relocating from uninhabitable areas, poverty alleviation, and engineering relocation in recent years, the public infrastructure has been gradually perfected, and most of the rural settlements are in the best position in the region. Studies have also shown that in recent years, the optimization of livability-related elements such as resources, ecology, architecture, etc. is increasingly being carried out around the region. Regional livability enhancements no longer depend solely on mass relocation. The increasing focus on the livability of rural settlements in the region is gradually focusing on certain individual livability indicators, such as increased security of settlement sites, improved transportation facilities, improved multi-functional utilization of resources, etc. Large-scale relocation works are not required, which is consistent with the findings of Bi, et al. [25].

From the perspective of the optimal mode of habitability of rural settlements in the upper reaches of the Minjiang River, this study divides the optimal mode of rural settlements into four categories, combining regional disaster constraints, humanistic characteristics, industrial development, etc., to promote the “life, production and ecological function” of the region. According to the relevant research by Yu, et al. [35], combined with the evaluation of the development suitability of the Jiuzhaigou earthquake-stricken area, it is best to optimize development zoning into tourism industry clusters, population clusters, agriculture and animal husbandry clusters and ecological protection zones. In addition, among all kinds of optimization modes, the agglomeration and promotion types focus on the construction of central villages and towns, while the demolition, reorganization and disaster prevention types pay attention to risk avoidance, the characteristic agriculture and animal husbandry development and integrated culture and tourism. Therefore, efforts are based on the regional industrial and cultural background to promote the upgrading of the regional industrial structure.

In conclusion, various livability problems exist in rural settlements, which not only are the main manifestations and main causes of inadequate rural development but also are closely related to the natural, socio-economic and cultural background in the region. This study is of positive significance for the improvement of the livability of rural settlements and the implementation of the beautiful countryside construction policy. At the same time, there are still deficiencies in this study, mainly reflected in the many factors affecting the livability of rural settlements. Although the evaluation index system for the habitability of rural settlements in the upper reaches of the Minjiang River is constructed from multiple perspectives, there are limitations in the selection of indicators, data acquisition, etc. Based on the screening of the main influencing factors and key indicators of rural settlements, future research will scientifically determine the rationality of the research process through model validation and field investigation and optimize it to improve the overall scientific and rational nature of the study.

## 5. Conclusions

The livability of rural settlements in the upper reaches of the Minjiang river is evaluated and optimized according to the evaluation results by using GIS technology and analyzing multiple datasets from the five aspects including site security, resource endowment, environmental suitability, convenience of life and economic development in rural settlements. The results show that:(1)In terms of individual livability, the upper reaches of the Minjiang river are prone to geological hazards and underdeveloped, and the site safety, resource endowment and economic affluence in the region are generally poor, while the environmental suitability and convenience of living are relatively good. The overall livability of the area is significantly better and the main factors affecting the livability are site security and economic prosperity.(2)In terms of the livability of rural settlements, the location of rural settlements was highly livability-oriented and the area of rural settlements in the study area decrease with the livability index, and the moderately and highly suitable areas are the core areas of human settlements.(3)Based on the results of the livability evaluation and field research, we summarize the problems faced by the livability improvement and propose four optimization models for the livability improvement of rural settlements based on the four optimization goals of development synergy, disaster management, industrial upgrading and cultural preservation.

## Figures and Tables

**Figure 1 ijerph-19-14712-f001:**
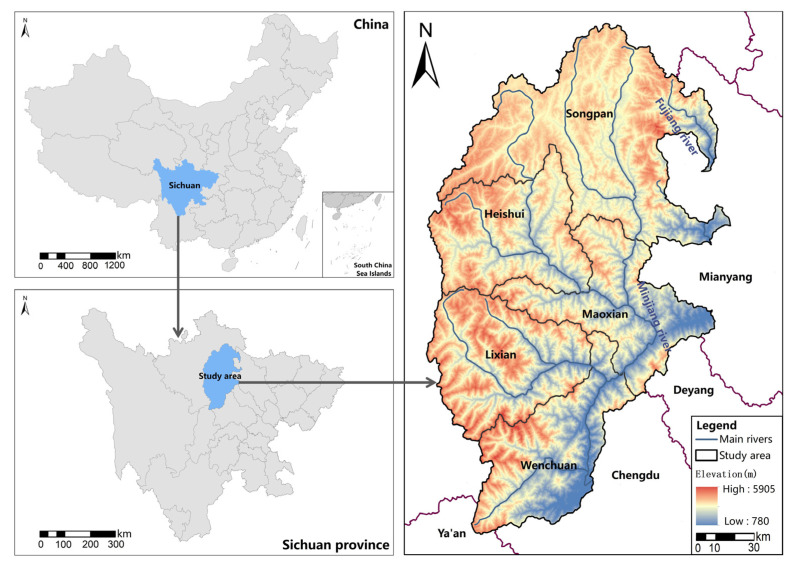
Location of the study area.

**Figure 2 ijerph-19-14712-f002:**
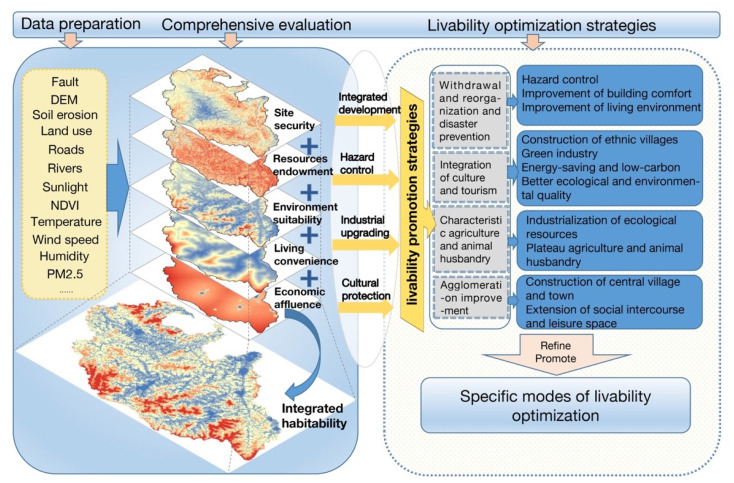
Evaluation and Optimization Framework of Rural Settlement Livability.

**Figure 3 ijerph-19-14712-f003:**
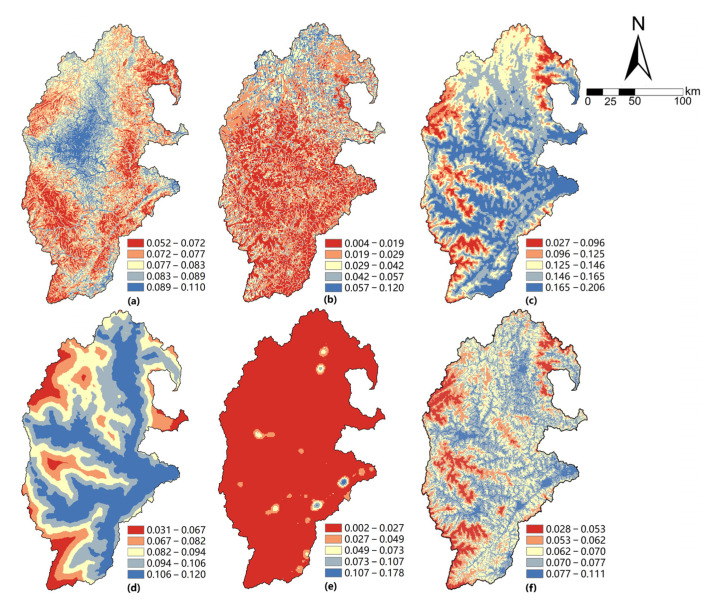
Individual Livability Classification Diagram in the Upper Reaches of the Minjiang River. (**a**) Site security; (**b**) Resources endowment; (**c**) Environmental suitability; (**d**) Living convenience; (**e**) Economic affluence; (**f**) Comprehensive livability.

**Figure 4 ijerph-19-14712-f004:**
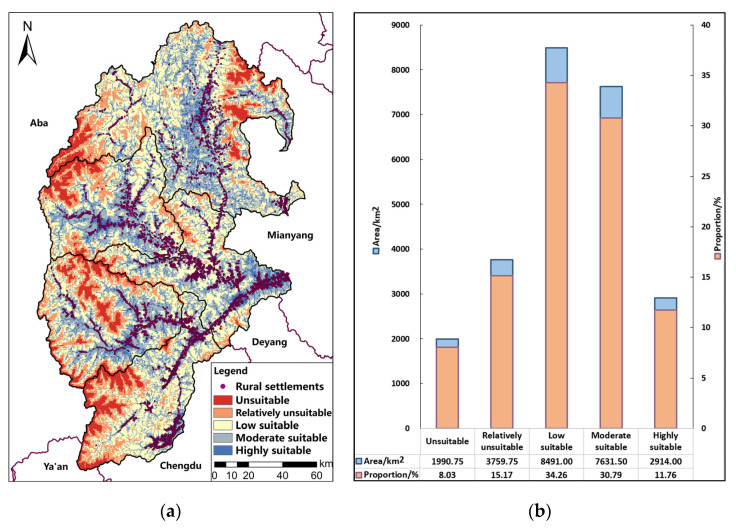
Livability of Rural Settlements in the Upper Reaches of the Minjiang River. (**a**) Comprehensive Livability Classification; (**b**) Livability Classification Statistics.

**Figure 5 ijerph-19-14712-f005:**
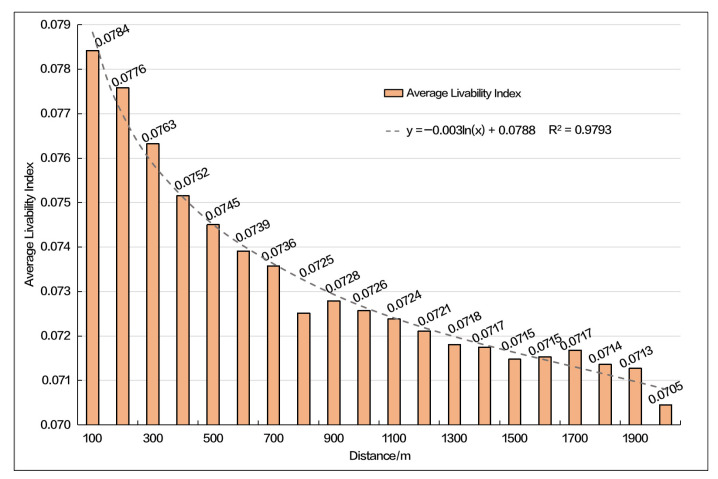
Average Livability Index in Different Buffers.

**Figure 6 ijerph-19-14712-f006:**
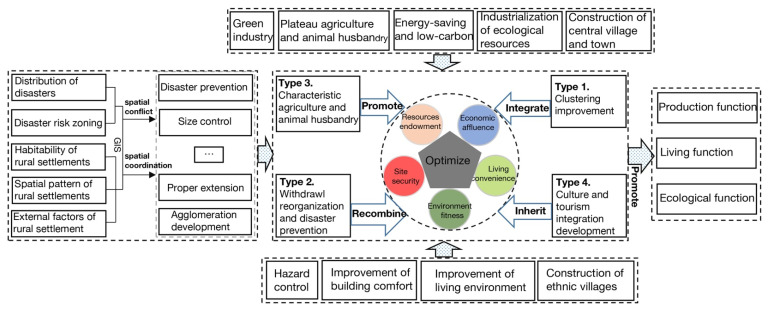
Optimization Model of Rural Settlement Livability in the Upper Reaches of the Minjiang River.

**Figure 7 ijerph-19-14712-f007:**
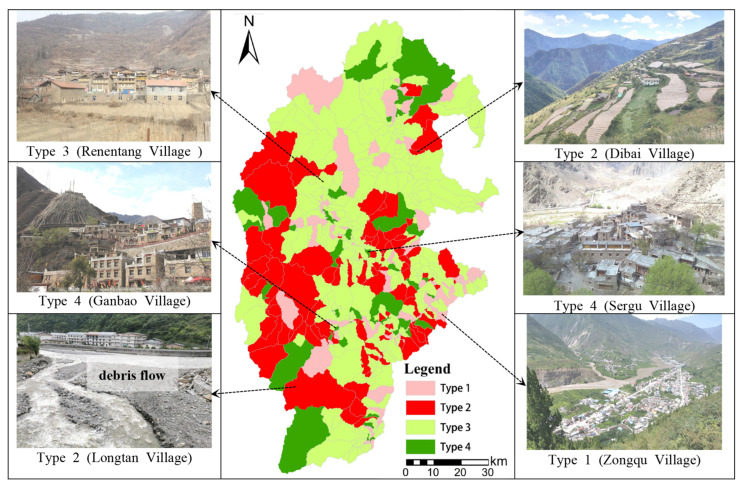
Zoning Diagram of Rural Settlement Optimization Model in the Upper Reaches of the Minjiang River.

**Table 1 ijerph-19-14712-t001:** Data Sources and Types.

The Data	Type	Resolution	Year	Data Source
Fault	Vector	-	-	https://www.eq-igl.ac.cn/, accessed on 20 July 2022.
Geological Disaster Points	Vector	-	2019	https://www.resdc.cn/, accessed on 2 March 2021.
DEM	Raster	30 m	-	https://srtm.csi.cgiar.org/srtmdata/, accessed on 21 March 2021.
Intensity of Soil Erosion	Raster	30 m	2020	https://www.resdc.cn/, accessed on 2 June 2021.
Land Use	Raster	10 m	2020	https://viewer.esaworldcover.org/, accessed on 24 June 2022.
Road Network	Vector	-	2020	http://www.openstreetmap.org/, accessed on 20 July 2022.
Water Network	Vector	-	2020	https://www.webmap.cn/, accessed on 20 July 2022.
Sunshine Duration	Excel	-	2020	https://www.resdc.cn/, accessed on 30 June 2022.
Soil Organic Carbon	Raster	1000 m	-	http://www.ncdc.ac.cn/, accessed on 24 June 2022.
NDVI	Raster	1000 m	2019	https://www.resdc.cn/, accessed on 23 June 2022.
Temperature	Raster	1000 m	2020	http://www.geodata.cn/, accessed on 22 June 2022.
Wind Speed	Raster	1000 m	2020	http://www.geodata.cn/, accessed on 23 June 2022.
Relative Humidity	Raster	1000 m	2020	http://www.geodata.cn/, accessed on 30 June 2022.
PM2.5 Concentrations	Raster	1000 m	2020	http://www.geodata.cn/, accessed on 22 June 2022.
Night Light Data	Raster	1000 m	2020	http://www.geodata.cn/, accessed on 23 June 2022.
POI	Vector	-	2020	https://lbs.amap.com/, accessed on 23 June 2022.
Spatial Distribution of Population	Raster	1000 m	2020	http://www.ornl.gov/sci/landscan/, accessed on 10 July 2022.
Spatial Distribution of GDP	Raster	1000 m	2019	https://www.resdc.cn/, accessed on 8 July 2022.
Rural Settlements	Vector	-	2018	Land-use Change Data for 2018
Administrative Boundaries	Vector	-	2021	https://www.webmap.cn/, accessed on 22 June 2022.

**Table 2 ijerph-19-14712-t002:** Evaluation Index System of Rural Settlement Livability.

The Target Layer	The System Layer	Indicator Layer/Positive and Negative Type	The Weight			References
			Entropy Method	AHP	Combined	
Evaluation of Livability of Rural Settlements	Site Security 0.1245	Distance from Fault (+)	0.02547	0.0185	0.0212	[36]
		Density of Geological Disaster Points (−)	0.00099	0.0151	0.0096	[37]
		Degree of Topographic Relief (−)	0.00102	0.0243	0.0152	[38]
		Slope (−)	0.00320	0.0359	0.0231	[33,35]
		Altitude (−)	0.00464	0.0457	0.0296	[36]
		Intensity of Soil Erosion (−)	0.00001	0.0427	0.0259	[29,39]
	Resource Endowment 0.3093	Grassland Area Per Capita (+)	0.06453	0.0099	0.0313	[40]
		Per Capita Arable Area (+)	0.13411	0.0206	0.0652	[40]
		Woodland Area Per Capita (+)	0.06149	0.006	0.0278	[40]
		Intensity of Road Network (+)	0.18073	0.0426	0.0968	[39]
		Intensity of Water Network (+)	0.12034	0.012	0.0545	[35,39]
		Sunshine Hours (+)	0.01317	0.0136	0.0134	[41,42]
		Soil Organic Matter (+)	0.01530	0.0233	0.0202	[29]
	Environmental Habitability 0.2311	Normalized Difference Vegetation Index (+)	0.00208	0.0809	0.0500	[31]
		Temperature and Humidity Index (+)	0.00298	0.1906	0.1169	[41,42]
		Wind Effect Index (+)	0.00320	0.0506	0.0320	[41,42]
		PM2.5 Concentration (−)	0.00371	0.0506	0.0322	[8]
	Accessibility of Life 0.1199	Road Accessibility (−)	0.00151	0.1213	0.0743	[43]
		Accessibility to Primary and Secondary Schools (−)	0.00147	0.0466	0.0289	[44]
		Accessibility to General Hospital (−)	0.00138	0.0267	0.0168	[45]
	Affluence of Economy 0.2152	Density of Population (+)	0.07981	0.0414	0.0565	[35,39]
		Level of GDP (+)	0.03986	0.0221	0.0291	[39]
		Night Light Index (+)	0.23067	0.0275	0.1073	[46]
		Distance from Factories (−)	0.00404	0.0114	0.0085	[35]
		Distance from Counties (−)	0.00429	0.0201	0.0139	[33,35]

**Table 3 ijerph-19-14712-t003:** Classification Statistics of Livability of Rural Settlements.

Classification of Habitability		Rural Settlement Areas	
		Area/km^2^	Proportion/%
Unlivable Areas	0.0277–0.0527	0.0000	0.000
Relatively Unlivable Areas	0.0527–0.0622	0.0105	0.018
Lower Habitable Areas	0.0622–0.0696	5.4208	9.309
Moderately Habitable Areas	0.0696–0.0771	21.4453	36.829
Highly Habitable Areas	0.0771–0.1106	31.3532	53.844
Sub-total	-	58.2298	100

## Data Availability

The data that support the findings of this study are available upon reasonable request from the authors.

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
