# Peer review of "Evaluation and Optimization Model of Rural Settlement Habitability in the Upper Reaches of the Minjiang River, China"

_ijerph, 2022, doi:10.3390/ijerph192214712_

Round 1
Reviewer 1 Report
Taking the upper reaches of the Min Jiang River as the research area, the paper constructs a rural settlement habitability evaluation index system, conducts research on habitability evaluation and optimization mode, and conducts field investigation and verification. And the research results provide certain theoretical support for the construction of beautiful and livable villages in the upper reaches of the Minjiang River and similar areas, which has good research value. The research is a significant work, but the paper needs very significant improvement before acceptance for publication. Detailed comments are as follows:
1.Title
(1)The Minjiang River is not as world-famous as the Amazon, the Yangtze River, or the Yellow River, so most people don't know that it's located there. The suggested title should be "Evaluation and Optimization Model of Rural Settlement Habitability in the Upper Reaches of Minjiang River, China".
2. Abstract:
(2) The abstract needs to have Background, Methods, Results, and Conclusion, and it needs to supplement the research methods used in the article. At the same time, the abstract needs to be further condensed to highlight the characteristics and innovation of the research.
3. Introduction
(3) The first paragraph needs to be further condensed, and only needs to explain the background and the significance of the research, indicating that the study is significant. Without over-discussing the relationship between rural revitalization and the habitability of rural settlements.
(4) The second paragraph describes the research related to habitability but does not focus on some of the research results related to the research in this paper, which needs to be supplemented.
(5) What are the characteristics and innovations of the research in this paper? And what is the point of merely duplicating existing work? If not, what new ideas have been proposed for the shortcomings or problems of certain studies? These need to be highlighted.
4. Materials and Methods
(6) The text of Figure 1 and Figure 2 is so tiny that the recognition is not good, which needs to be adjusted; In addition, the legend and scale bar needs to be adjusted, and the relevant contents are also modified according to the requirements of the picture.
(7) “For the sake of harmonizing the research standards and improving the research accuracy, this paper uniformly resamps the acquired data to the grid unit of 500m×500m in ArcGIS 10.2 platform”, Why resampling to 500m×500m, and what is the purpose? What is the basis? Is it based on the relevant research results, or is it based on experimental comparison?
At the same time, the spatial resolution of the data previously used (such as Relative Humidit, Night Light Data, Spatial Distribution of GDP) is 1km×1km, how to reflect and ensure the accuracy of the data?
(8) The calculation of index weights is related to the accuracy of evaluation results. The article mentions that the AHP-entropy method combination is used to determine the index weight, and the combined coefficient of the α and the β objective weight is calculated, why these methods are used? And what is the result?
5. Results
(9)Figure 4(a) is not recognizable, you need to adjust the legend, scale bar, and text size; Figure 4(b) is ill-cartographed, with missing horizontal coordinates and unclear text.
(10) "There are 17,492 rural settlements in the upper reaches of Minjiang River...", the article uses village and settlement map spots for statistical and optimization model classification, so the number of villages and settlements in the upper reaches of the Minjiang River should be appropriately elaborated in the regional overview.
(11) The connection between the rural settlement optimization model in the article and the previous evaluation results is insufficient. Because the evaluation system is established from five aspects, habitability is divided into five levels. So My suggestion is to optimize the evaluation results, especially in those areas with poor livability should be the focus of optimization areas. But now the article is divided into four modes, and it is difficult to directly observe the connection with the habitability evaluation results, and why?
6. Discussion and Conclusions
(12) During the discussion, it is necessary to explain what is the contribution of this paper's research to relevant issues in the world and regional development, which will be more conducive to the dissemination of literature and reflect the value of the article.
(13) “...uses GIS spatial analysis technology to explore the suitability of regional rural settlements.” The methods description is not specific enough to be summarized by remote sensing and GIS only.
Author Response
Response to Reviewer 1 Comments
Point 1: The Minjiang River is not as world-famous as the Amazon, the Yangtze River, or the Yellow River, so most people don't know that it's located there. The suggested title should be "Evaluation and Optimization Model of Rural Settlement Habitability in the Upper Reaches of Minjiang River, China".
Response 1: Thank you for the reviewer's suggestion, you have made a good point and we have revised the title of the article to "Evaluation and Optimization Model of Rural Settlement Habitability in the Upper Reaches of Minjiang River, China".
Point 2: The abstract needs to have Background, Methods, Results, and Conclusion, and it needs to supplement the research methods used in the article. At the same time, the abstract needs to be further condensed to highlight the characteristics and innovation of the research.
Response 2: A description of the study's research methodology has been added and the Abstract has been revised and improved as suggusted by the reviewer.
Point 3: The first paragraph needs to be further condensed, and only needs to explain the background and the significance of the research, indicating that the study is significant. Without over-discussing the relationship between rural revitalization and the habitability of rural settlements.
Response 3: The first paragraph has been further condensed as suggested by the reviewer, and the research background and significance of the study have been added and improved.
Point 4: The second paragraph describes the research related to habitability but does not focus on some of the research results related to the research in this paper, which needs to be supplemented.
Response 4: Thank you for your suggestion, we have revised and improved the introduction section and added research results related to this study in the second and third paragraph.
Point 5: What are the characteristics and innovations of the research in this paper? And what is the point of merely duplicating existing work? If not, what new ideas have been proposed for the shortcomings or problems of certain studies? These need to be highlighted.
Response 5: Thank you for your suggestion. The main innovation of this study lies in the selection of the study area and the construction of the livability evaluation index system. At present, most of the studies on settlement livability evaluation and optimization models focus on urban settlements, but there are very few research results on rural settlements, especially those in geological disaster-prone areas. The upper reaches of Minjiang River is ecologically fragile, disaster-prone and economically backward, and the combination of these factors has led to a more complex evolution of rural settlements and a more prominent livability problem. In addition, this paper also takes the regional geological hazards, ecological environment and socio-economic conditions into account in the construction of the livability evaluation index system, and innovatively constructs the livability evaluation index system from five aspects: site safety, resource endowment, environmental suitability, convenient living and economic affluence, spatializing the per capita resource factors, and exploring the comprehensive livability status of the region. The relevant descriptions are highlighted in the last paragraph of the introduction section.
Point 6: The text of Figure 1 and Figure 2 is so tiny that the recognition is not good, which needs to be adjusted; In addition, the legend and scale bar needs to be adjusted, and the relevant contents are also modified according to the requirements of the picture.
Response 6: We thank you for your suggestion, and have checked and modified the legends, scales, and displays of all the images in this paper, and output higher pixel precision images to improve the recognition effect.
Point 7: “For the sake of harmonizing the research standards and improving the research accuracy, this paper uniformly resamples the acquired data to the grid unit of 500m×500m in ArcGIS 10.2 platform”, Why resampling to 500m×500m, and what is the purpose? What is the basis? Is it based on the relevant research results, or is it based on experimental comparison?At the same time, the spatial resolution of the data previously used (such as Relative Humidit, Night Light Data, Spatial Distribution of GDP) is 1km×1km, how to reflect and ensure the accuracy of the data?
Response 7: The 500m-sized grid unit was determined as a result of literature review, data processing and field verification. On the one hand, based on the research results of Yu, Li, and He, etc., it is considered that the 500m-sized grid unit is more reasonable in the upper reaches of Minjiang River. Due to the complex and variable geological and geomorphological conditions, disasters and concentrated economic factors in the region, the 1000m-sized grid unit is too large in comparison, which may lead to the situation that the overall accuracy of the evaluation is not so good. On the other hand, in the data processing stage, this study tried to calculate and evaluate based on 100m, 500m, and 1000m-sized grid units respectively, and the results showed that the evaluation results of 500m-sized grid unit were closest to the reality, while the evaluation results of the 1000m-sized grid unit were slightly rough, and the 100m-sized grid cell could not guarantee the accuracy of the acquired and used data. Therefore, this paper chooses to use the 500m×500m size grid unit for the evaluation of the livability of rural settlements. As suggested by the reviewer, the reasons for choosing 500m×500m grid unit have been added in the relevant part of the paper.
Point 8: The calculation of index weights is related to the accuracy of evaluation results. The article mentions that the AHP-entropy method combination is used to determine the index weight, and the combined coefficient of the α and the β objective weight is calculated, why these methods are used? And what is the result?
Response 8: The AHP-Entropy method was used to assign weights to each factor, of which the AHP method is a subjective assignment method with strong interpretation. AHP (subjective) combined with the Entropy (objective) weighting method can obtain weights that are both objective and consistent with actual empirical judgments. However, due to the large amount of data within the study area, in order to eliminate the interference of large fluctuating data and make the degree of difference between subjective and objective weights consistent with the degree of difference between α and β, the concept of distance function is introduced by drawing on relevant studies, and the calculation results show that α and β are 0.6074 and 0.3926 respectively, and the relevant results and references are clearly marked in the 2.4.6 Index weight.
Point 9: Figure 4(a) is not recognizable, you need to adjust the legend, scale bar, and text size; Figure 4(b) is ill-cartographed, with missing horizontal coordinates and unclear text.
Response 9: Thanks to the reviewer's suggestion, the legend, scale and display of Figure 4(a) in the text have been checked and revised and replaced. And the text of 4(b) has been adjusted, with the horizontal axis marked as the area and percentage of livability zoning.
Point 10: "There are 17,492 rural settlements in the upper reaches of Minjiang River...", the article uses village and settlement map spots for statistical and optimization model classification, so the number of villages and settlements in the upper reaches of the Minjiang River should be appropriately elaborated in the regional overview.
Response 10: Thanks to the reviewer's suggestion, this study involves a total of 17,492 rural settlements and that the subsequent evaluation and enhancement of the livability of rural settlements is based on this, a description of the number and area of rural settlements has been included in the 2.1 study area section as suggested.
Point 11: The connection between the rural settlement optimization model in the article and the previous evaluation results is insufficient. Because the evaluation system is established from five aspects, habitability is divided into five levels. So My suggestion is to optimize the evaluation results, especially in those areas with poor livability should be the focus of optimization areas. But now the article is divided into four modes, and it is difficult to directly observe the connection with the habitability evaluation results, and why?
Response 11: Thanks to the reviewer's suggestion. In this study, the model for improving the livability of rural settlements is proposed in accordance with the steps of livability evaluation - problem summary - optimization objectives - specific model. Based on the evaluation results of the livability of rural settlements and field research, five major problems are conducted from the perspective of site safety, resource endowment, environmental suitability, convenient living and economic affluence. Firstly, the rural settlements near the county and the core township often only carry the residence function, and the internal force of development is insufficient. Secondly, some rural settlements are too far from the centre of the settlement, scattered, and mostly inhabited by the elderly or directly abandoned, such as Sandagu Village in Heshui County. Thirdly, the safety of some rural settlements is not guaranteed or the safety hazards are high, such as Baibu Village in Maoxian County and Zhanpai Village in Wenchuan County. Fourth, the overall resource condition of the region is poor and underutilized, and the industrial support for regional development is insufficient. Fifth, the lack of preservation and excavation of cultural genes in some rural settlements leads to the loss of cultural characteristics. Based on the five major problems above, We summarize the four optimization goals: development synergy creation, disaster management, industrial upgrading and cultural preservation, and propose four optimization models: Clustering Improvement, Restructuring and Disaster Prevention, Agriculture and Animal Husbandry Clustering Improvement, Restructuring and Disaster Prevention, Agriculture and Animal Husbandry Development, and Integrated Development of Culture and Tourism. The relevant contents have been added in section 3.3 of the paper.
Point 12: During the discussion, it is necessary to explain what is the contribution of this paper's research to relevant issues in the world and regional development, which will be more conducive to the dissemination of literature and reflect the value of the article.
Response 12: Thanks to the reviewer's suggestion. The contribution of this study is to provide clear ideas and scientific basis for enhancing the livability of rural settlements in the upper Minjiang River and similar disaster-prone areas in the world, and to enrich and optimize the theoretical framework for the study of livability evaluation and enhancement strategies of rural settlements in disaster-prone areas. The value and significance of this study for the world and regional development have been added in the first paragraph of the discussion section.
Point 13: “...uses GIS spatial analysis technology to explore the suitability of regional rural settlements.” The methods description is not specific enough to be summarized by remote sensing and GIS only.
Response 13: Thanks to the reviewer's suggestion, the research method of this paper is mainly based on GIS spatial analysis, kernel densitometry analysis, and minimum cumulative resistance model, etc. The spatial superposition method of grid data is used to transfer each evaluation index to the same spatial base, and the livability index corresponding to each grid is obtained with the help of map algebra operation, and finally the livability classification of rural settlements in the upper Minjiang River is realized. A detailed description of the study methodology has been added to the discussion section of this paper.

Reviewer 2 Report
Dear Authors,
I read with great interest your article entitled Evaluation and Optimization Model of Rural Settlement Habitability in the Upper Reaches of Minjiang River. I appreciate your critical view of a very complex issue and your attempt to frame not only the quantitative but also the qualitative aspects in a mathematical and planning framework. Unfortunately, in my opinion, to be able to gain the interest wider range of researchers, you need to present it in a little more detail, outline the background for readers outside China and East Asia, and set the results in a broader context in a way that allows the research to be replicated and verified. Furthermore, being aware of the limitations in the representativeness of the results achieved as expressed, rightly, in lines 504-510, you should not build on their strategies and models, particularly those assuming the relocation of thousands of people (cf. lines 400-403 and Figure 7: Type 2 Rural Settlement Optimization Model).
I encourage you to submit a revised version of the manuscript taking into account, among other things, the comments below.
General comments:
-
It is always risky to use words such as 'beautiful' (lines 29, 46, 61, 502) while referring to something based on statistics or mathematic models and presented on a territorial scale (Zhang [6] uses the term 'amenity’, which is similar, but not the same) or 'natural’ while referring to rural, i.e., heavily transformed anthropic landscape. Also, consider revising terms such: as ecological relocation, construction of ethnical villages, risk aversion (if not applying to economics and finance or psychology), the river's upper reaches instead of the upper river basin, etc.
-
The manuscript is suffering from some contradictions: from the overall evaluation of the livability of the region (cf. lines 19-21 and 300-301, etc.) to the universality of the case study (cf. line 116 where you describe the upper Minjiang River basin as typical to other alpine and canyon areas and line 132 where you reveal its equatorial characteristics (yellow soil), etc.) and beyond.
-
In mathematic models and multi-criteria analysis, it’s all about weight. How it is estimated?
-
Why are primary indexes composed of 7, 5, 4, or 3 secondary indexes? How they were defined? Why population is a separate index as well as a merged factor (secondary indexes calculated per capita)? Why typical risk indicators are not taken into consideration? Are there no other threats to be included, for instance, those related to climate change? Floods? Droughts? Heat waves? Non-typical weather phenomena?
-
Section 2.4 - each index and each secondary index should be properly described.
-
Section 3. Thresholds should be defined.
-
Are you sure you can build models and strategies on the results obtained without corroborating them first? For instance, calculating the impact (or the result) of the optimization proposed. Or estimating the time, money, and effort necessary for the application?
-
Why 5 indexes and 4 types? Optimization Model of Rural Settlements Livability seems something applied axiomatically but is not presented as such.
-
Most of the abstract as well as the conclusions are redundant. Correlation between factors and indexes would be more interesting to readers than percentages.
Details & edition:
- The first two sentences are redundant.
- Some references are lacking, for instance: Line 38 - which reform are you referring to? Line 58 - which WHO document are you referring to? Line 467 - it is not resulting from your research, a reference is needed, etc.
- Line 41-42: only farmers are aging? Are all rural residents farmers?
- Line 96: consider opening a new section.
- Reading is hampered by linguistic errors, including syntax and missing or doubled words, inappropriate terminology, and punctuation errors ranging from simple typos and double spaces to inappropriate use of inverted commas (in particular lines 49-50, 77-78, 102-108, 428, 432-436, 457, and others). Proofreading by a native speaker is recommended.
I hope the above comments prove inspirational. Their sole purpose was to point you toward the best possible version of your paper so your research could gain the attention and appreciation it deserves.
Best regards and all the luck.
Reviewer 3 Report
Literature review can be improved
Figure 3
An additional figure where a 500x500 grid is shown to explain better how single parameters work together to understand liveability index.
In the left-lower figure it is difficult to locate rural settlment, maybe a vectorial representation is more suitable, for example dots.
Table 1:
Because most of the geodata have a 1000 meters precision, it would advisable to use a similar size grid.
These data:
DEM - Intensity of Soil Erosion - Land Use have a 30 meteres precision. These level of precision is unuseful because it adds complexity in the whole calculation which. A lower precision geodatabase would be advisable.
line:279. Because your case study is in a single area of China the following statement should be justified:
The north is significantly better than the south, 279 which is related to the per capita occupancy of woodland, arable land and grassland resources in the north.
Author Response
Response to Reviewer 3 Comments
Point 1: Literature review can be improved.
Response 1: Thanks to the reviewers' suggestion, we have revised and improved the introduction section. The first paragraph has been condensed to clarify the background and relevance of the study; the second and third paragraphs have been added with some specific descriptions of related studies; the fourth paragraph has been revised to better highlight the innovation and relevance of the study.
Point 2: Figure 3,An additional figure where a 500x500 grid is shown to explain better how single parameters work together to understand liveability index.
Response 2: Thanks to the reviewers' suggestion, your consideration is valid and we have added the results of the comprehensive livability assessment of the study area shown in the 500x500 grid in Figure 3, added the relevant legend for ease of reading and replaced the image.
Point 3: In the left-lower figure it is difficult to locate rural settlement, maybe a vectorial representation is more suitable, for example dots.
Response 3: Thanks to the reviewers' suggestion, the images have been revised, the legend and scale have been optimized, and the rural settlement patches have been replaced with rural settlement points as suggested by the reviewer.
Point 4: Table 1: Because most of the geodata have a 1000 meters precision, it would advisable to use a similar size grid.
Response 4: Thanks to the reviewers' suggestion, the 500m-sized grid unit was determined as a result of literature review, data processing and field verification. On the one hand, based on the research results of Yu, Li, and He, etc., it is considered that the 500m-sized grid unit is more reasonable in the upper reaches of Minjiang River. Due to the complex and variable geological and geomorphological conditions, disasters and concentrated economic factors in the region, the 1000m-sized grid unit is too large in comparison, which may lead to the situation that the overall accuracy of the evaluation is not so good. On the other hand, in the data processing stage, this study tried to calculate and evaluate based on 100m, 500m, and 1000m-sized grid units respectively, and the results showed that the evaluation results of 500m-sized grid unit were closest to the reality, while the evaluation results of the 1000m-sized grid unit were slightly rough, and the 100m-sized grid cell could not guarantee the accuracy of the acquired and used data. Therefore, this paper chooses to use the 500m×500m size grid unit for the evaluation of the livability of rural settlements. As suggested by the reviewer, the reasons for choosing 500m×500m grid unit have been added in the relevant part of the paper.
Point 5: These data: DEM - Intensity of Soil Erosion - Land Use have a 30 meteres precision. These level of precision is unuseful because it adds complexity in the whole calculation which. A lower precision geodatabase would be advisable.
Response 5: The main reason for selecting these data is that the study area is located in a high mountain valley area, the topography is more complex in the region, and the spatial differences in dem and soil erosion intensity are more obvious, and also for more accurate grasp of the per capita resource occupation in the region. In addition, in the subsequent calculations, we also unified the data to a 500mx500m grid, and the evaluation results were verified and modified by field surveys until they reached the optimum.
Point 6: line:279. Because your case study is in a single area of China the following statement should be justified:The north is significantly better than the south, 279 which is related to the per capita occupancy of woodland, arable land and grassland resources in the north.
Response 6: Thanks to the reviewer's suggestion, there seems to be a certain lack of clarity of expression, and we have revised the paragraph in the text.

Round 2
Reviewer 2 Report
Dear Authors,
I appreciate the effort you put in rewriting your paper in such a short time. If I may, I would insist on expanding the background. If the construction of beautiful villages (or any other policy you referred to) is a specific policy in China, it should be presented as such. If I were you, I would re-check the manuscript with that in mind. Whatever is obvious to you may not be obvious to readers from different countries and different fields.
Best regards.
Author Response
Dear Reviewer,
Thank you very much for taking your time to check our manuscript and give us suggestions. Those comments are all valuable and helpful for revising and improving our paper, and we have revised the paper according to your suggestions, which makes our paper more reasonable. In addition, based on your suggestion (Round 2), we have addressed the policy description in the paper, and expanded the background through an international perspective.
Thanks again and Best regards.